# Plastic Antibody of Polypyrrole/Multiwall Carbon Nanotubes on Screen-Printed Electrodes for Cystatin C Detection

**DOI:** 10.3390/bios11060175

**Published:** 2021-05-31

**Authors:** Rui S. Gomes, Blanca Azucena Gomez-Rodríguez, Ruben Fernandes, M. Goreti F. Sales, Felismina T. C. Moreira, Rosa F. Dutra

**Affiliations:** 1BioMark@ISEP, School of Engineering, Polytechnic Institute of Porto, 4249-015 Porto, Portugal; deb12008@fe.up.pt (R.S.G.); goreti.sales@eq.uc.pt (M.G.F.S.); 2BioMark@UC, Department of Chemical Engineering, Faculty of Science and Technology, University Coimbra, 3030-790 Coimbra, Portugal; 3CEB, Centre of Biological Engineering, University of Minho, 4715-000 Braga, Portugal; 4Biomedical Engineering Laboratory, Federal University of Pernambuco, Recife-PE 50670-901, Brazil; blanca.gr@teziutlan.tecnm.mx; 5LaBMI–PORTIC, Laboratory of Medical and Industrial Biotechnology–Porto Research, Technology & Innovation Centre, Polytechnic Institute of Porto, 4200-472 Porto, Portugal; rfernandes@ess.ipp.pt; 6Escola Superior de Saúde, Polytechnic Institute of Porto, 4200-072 Porto, Portugal; 7i3S-Institute of Health and Research Innovation, University of Porto, 4200-135 Porto, Portugal

**Keywords:** cystatin C, molecularly imprinted polymer, electrochemical biosensor, polypyrene, multiwall carbon nanotubes, acute kidney injury

## Abstract

This work reports the design of a novel plastic antibody for cystatin C (Cys-C), an acute kidney injury biomarker, and its application in point-of-care (PoC) testing. The synthetic antibody was obtained by tailoring a molecularly imprinted polymer (MIP) on a carbon screen-printed electrode (SPE). The MIP was obtained by electropolymerizing pyrrole (Py) with carboxylated Py (Py-COOH) in the presence of Cys-C and multiwall carbon nanotubes (MWCNTs). Cys-C was removed from the molecularly imprinted poly(Py) matrix (MPPy) by urea treatment. As a control, a non-imprinted poly(Py) matrix (NPPy) was obtained by the same procedure, but without Cys-C. The assembly of the MIP material was evaluated in situ by Raman spectroscopy and the binding ability of Cys-C was evaluated by the cyclic voltammetry (CV) and differential pulse voltammetry (DPV) electrochemical techniques. The MIP sensor responses were measured by the DPV anodic peaks obtained in the presence of ferro/ferricyanide. The peak currents decreased linearly from 0.5 to 20.0 ng/mL of Cys-C at each 20 min successive incubation and a limit of detection below 0.5 ng/mL was obtained at pH 6.0. The MPPy/SPE was used to analyze Cys-C in spiked serum samples, showing recoveries <3%. This device showed promising features in terms of simplicity, cost and sensitivity for acute kidney injury diagnosis at the point of care.

## 1. Introduction

Worldwide, chronic kidney disease (CKD) constitutes a great economic impact with high rates of morbidity and mortality [1]. Renal substitutive therapy is a burden, especially for low-income countries, since their socio-economic conditions restrain effective programs of non-preventable cardiovascular diseases and diabetes, drug therapy, tobacco control, promotion of physical activity and the reduction of salt intake through legislation and food [2]. Patients undergoing chronic renal replacement therapy have an increased chance of acute kidney injury (AKI) and consequent death [3]. AKI is characterized by a rapid decline in the glomerular filtration rate, thus, biomarkers should be continuously monitored in chronic renal patients to increase their survival rates [3,4]. Cystatin C (Cys-C) is a single non-glycosylated polypeptide chain consisting of 120 amino acid residues with a molecular mass of 13 kDa, and is more specific as a renal biomarker for glomerular filtration than creatinine, because it does not depend on gender, age, diet and muscle mass [4,5,6,7,8]. 

Thus, development of new methods for monitoring Cys-C that are user-friendly and practical in point-of-care (PoC) settings could represent a strategy to follow patients with AKI. Due to the major advances in nanotechnology [9], several biosensors for Cys-C have been developed for this purpose. Most of these are immunosensors and employ naturally derived antibodies. They include impedimetric systems with an interdigitated electrode [10] or a three-electrode system [11], spectroelectrochemical systems using TiO_2_ nanotube arrays [12] or gold nanopillar substrates [13], electrochemiluminescence systems using Au/Pd/Pt nanoflowers modified with MoS2 nanosheets [14] or a graphene sheet modified with a layer of rubrene [15] or reflectometric interference spectroscopy systems employing glass substrates [16]. As an alternative to immunosensors, enzymatic-based sensors have also been developed [17]. Overall, all these sensing materials are of natural origin, making the devices less stable, less reproducible and more expensive.

Alternative synthetic recognition materials include molecularly imprinted polymers (MIPs), working as biorecognition elements that ensure that electrochemical responses come from Cys-C. MIPs are known as plastic antibodies for mimicking the behavior of antibodies obtained from biological sources. These polymers can selectively recognize a given target molecule to which they are designed. If applied as recognition units of biosensors, these receptors provide very high selectivity, so the use of MIPs as recognition units in biochemical sensors is gaining increasing interest. Overall, the presence of a biorecognition element is essential, or else the response of the biosensor to real samples would be linked to all compounds present in the sample and not only to Cys-C, as the working electrode would absorb/adsorb any protein/biomolecule present. Thus far, there is no MIP material for Cys-C that we know of. In general, MIP materials offer long-term stability, are inexpensive and can be tailor-made on demand, for almost any intended target compound [18,19]. Moreira et al. (2013) are among the pioneers of this field, proving that the integration of plastic antibodies for detecting protein biomarkers in simple biosensing technology is possible [20]. MIP materials may be tuned to display conductive features, fulfilling electron transfer requirements and benefiting the subsequent electrochemical responses. 

MIP materials may be composed of conducting polymers, such as polypyrrole (PPy), a widely known conducting polymer that could be employed as a polymeric network of MIP material, offering easy electropolymerization [21]. In combination with carbon nanotubes (CNTs), the nanocomposite PPy/CNT exhibits a high doping and de-doping rate and a high capacity of charge storage, making it a supercapacitor [22]. When a PPy/CNT composite is obtained by electrosynthesis, it achieves a long cycle life, derived from strong π–π bonds between the PPy conjugated structure and the CNT [10].

Thus, this work reports, for the first time, MIP material for Cys-C, produced by electropolymerizing in situ Py-based monomers in the presence of the target protein. The contribution of additional conductive materials based on CNTs was also evaluated. The MIP synthesis was optimized, characterized and suitable and applied to check application feasibility of the final biosensor. 

## 2. Experimental Section

### 2.1. Apparatus

The electrochemical measurements were conducted with a potentiostat/galvanostat from Metrohm Autolab and a PGSTAT302N. Carbon SPEs (C-SPEs) were from Dropsens (DRP-C110, Oviedo, Spain). The working electrode had a diameter of 4 mm and working/counter electrodes were made of carbon and the reference electrode was made of silver. C-SPEs were interfaced with the potentiostat via a specially designed switch box (BioTID, Porto, Portugal). Raman spectroscopy data were collected by Thermo Scientific DXR equipment, with a confocal microscope and a 532 nm laser. The Raman spectrometer was operated with 2 mW laser power and a 50 μm slit aperture.

### 2.2. Reagents and Solutions

De-ionized laboratory grade water was employed, and all chemicals were of analytical grade. Potassium hexacyanoferrate III (K_3_[Fe(CN)_6_]), potassium hexacyanoferrate II (K_4_[Fe(CN)_6_]) trihydrate, L-ascorbic acid and sodium acetate were obtained from Riedel-de Häen. Cys-C, MWCNTs and urea were obtained from Fluka. Py, sodium chloride (KCl) and Py-COOH (Py-3-carboxylic acid, 99%) were obtained from Merck. Creatine kinase iso-enzyme was obtained from European Reference Materials. Creatinine and bovine serum albumin were obtained from Amresco. 

Electrochemical readings were carried out in 5.0 × 10^−3^ mol/L K_3_[Fe(CN)_6_] and K_4_[Fe(CN)_6_] redox standard solution, prepared in 0.1 mol/L KCl. Selectivity studies used synthetic serum spiked with other compounds that could act as interfering species (creatine kinase-MB 0.2 g/L, ascorbic acid 0.15 g/L, creatinine 1g/L and bovine serum albumin 12 g/L). Spiked serum samples were diluted 10 times in human serum obtained as Cormay® HN. 

### 2.3. Electrochemical Procedures

Electrochemical data were obtained by cyclic voltammetry (CV) and differential pulse voltammetry (DPV). For the control of the surface immobilization, CV was carried out from −0.6 to +0.6 V with a scan rate of 20 mV/s and DPV was carried out from −0.3 to +0.3 V. Electrochemical readings were obtained for MPPy and NPPy materials using a minimum of three replicate readings (*n* < 3). Calibration curves used DPV data using the NOVA software program.

The calibration curve was made by incubating increasing concentrations of Cys-C standard solutions for 20 min. After each concentration, the electrochemical response of the standard probe [Fe(CN)_6_]^3−^/^4−^ was collected, obtaining in this stage the electrical features. The Cys-C concentrations ranged from 0.5 to 40 ng/mL, prepared in acetate buffer pH 6.0.

Selectivity data were collected by incubating Cys-C standard solutions prepared with diluted spiked Cormay^®^ serum. 

### 2.4. Production of the Plastic Antibody on the C-SPE

C-SPEs were first pre-treated by applying +1.7 V for 200 s to a 0.1M KCl solution. The MPPy film was obtained as described in Figure 1. Electropolymerization of both MPPy and NPPy was achieved by 10 CV scans, with a start potential of −0.5 V, a lower vertex potential of −0.8 V and an upper vertex potential +0.8 V, with a scan rate of 20 mV/s. The MPPy material was obtained in a pH 6 acetate buffer solution containing MWCNTs (40 %), Py (80.0 mol/L), 10% Py-COOH (4.0 mmol/L) and 1% Cys-C (0.050 µg/mL). The NPPy material was obtained by the same procedure, but without Cys-C in the solution. 

The use of pH 6 and 1% Cys-C followed previous preliminary experiments involving other protein imprinting assemblies. The overall composition was selected according to the experience of the research groups and optimization steps. In addition, the use of a small amount of Py-COOH compared to Py followed the same principle as that of the preparation of a material denoted as SPAM in the literature [23]. In it, the protein is surrounded by a monomer that is different from the overall polymeric matrix, aiming to enhance the capacity of recognizing this protein (it has a higher affinity to the binding site). 

Cys-C from was extracted from the polymeric network by incubating the working electrode area of the MPPy/C-SPEs in 0.1M urea for 3 h. The electrode surface was then washed several times in acetate buffer to remove any contaminants on the surface and rinsed with water.

## 3. Results and Discussion

### 3.1. Follow-Up of the Surface Modification

The Raman spectra of carbon working electrodes on the C-SPE and their subsequent modification with MPPy and NPPy materials are shown in Figure 2. As expected from the literature, the G- and D-bands are the more relevant peaks [24,25]. The G-band shows the presence of sp^2^ hybrid orbitals in rings and chains, while the D-band reveals hexagonal lattice defects of carbon-based materials, including sp^3^–carbon hybridization. The G- and D-bands of the pre-treated C-SPE were at 1577.6 cm^−1^ and 1314.2 cm^−1^ Raman shifts [25], and those of the MPPy and NPPy materials moved to higher Raman shift values, thereby confirming the modification made to the substrate. It may be that the modification consisted of the formation of a thin film of PPy with CNTs and with carboxylated moieties from the Py-COOH, because the spectra obtained had no evidence of specific PPy peaks. 

Moreover, the changes in the ratio of the intensity of G- and D-bands was analyzed, as they reflect changes in the organization of the carbon materials and may help to confirm the chemical modifications made to the carbon electrode. In simple terms, an increasing ID/IG ratio reveals increasing disorder degrees within the materials [26]. The ID/IG ratio of the working electrode of non-modified C-SPE was 0.87, while the ratios of MPPy and NPPy were 0.72 and 0.67, respectively. These decreasing ratios on the polymer-modified electrodes confirmed the presence of a polymeric film because PPy is a conducting polymer with conjugated C=C bonds and CNTs were introduced in the polymeric network. The fact that NPPy had a lower ratio than MPPy was related to two effects: (1) the non-imprinted film does not contain Cys-C molecules entrapped within the network, because they were absent in the synthesis process; (2) the film may have been formed to a greater extent, because the presence of a protein at the time of polymer growth hinders polymer formation.

Overall, Raman analysis confirmed the presence of the polymeric materials on both MPPy and NPPy devices and the differences between imprinted and non-imprinted devices seem to confirm a different composition of the sensing materials.

### 3.2. Polypyrrole-Based Sensing Element

A PPy-based material was employed as the basis for creating a sensing element of Cys-C, due to its excellent conductivity features that may enhance sensitivity. This was done by electrochemical polymerization, in which the oxidation of the monomers under a suitable anodic potential or current produces a radical cation. These radical species are highly unstable, reacting with each other to create a radical dimer, which in turn is transformed into a trimmer, leading, in time, a to longer polymer chain. Apart from being able to produce a 3D network capable of recognizing Cys-C, the resulting polymer should lead to a stable electrochemical signal. This is fundamental to ensuring that the electrochemical signal changes are related only to Cys-C and not to random background changes or signal drifting. This depends mostly on the conditions selected to obtain the polymeric network. Thus, different NPPy devices were prepared first by CV, for optimizing the production of this polymeric film. 

The effect of the potential range of the CV scan was evaluated first, as it contributes to the way the polymer grows, keeping in mind that the oxidation peak of the monomer Py is included within this range. Three different potential ranges were tested: (i) −0.80 to +0.65 V, (ii) −0.80 to +0.80 V and (iii) −0.80 to +1.40 V. In general, the increasing potentials increased the number of radical species formed and thereby the extent of polymer formation, as more energy was being introduced into the electrochemical system. While the higher potential values had the possibility of leading to the overoxidation of the polymeric film, the lower values displayed poorer reproducibility. This poor reproducibility reflected the fact that the voltage required to reach the maximum Py oxidation (maximum current) was far from being achieved. Thus, the selected range of potential was within −0.80 to +0.80V, which ensured the production of a reproducible and stable polymeric layer.

The stability of the NPPy film formed is also intrinsically linked to the potential range selected, according to our experience. A total of 10 CV scans was used in each condition, with a scan rate of 20 mV/s. In general, the number of selected CV scans is based on the stability of the sensor after successive readings with a redox probe and washing steps with the buffer. Using 10 CV scans, the sensor displayed reproducible and stable electrochemical data.

The behavior of the resulting films was evaluated by CV and DPV readings in the presence of the standard redox probe solution. The data obtained are shown in Figure 3. When the films were obtained in the potential range −0.80 to +0.65 V, the peak separation was lower than 0.15V, with the redox probe showing a quasi-reversible behavior. However, the successive incubations of the acetate buffer solution yielded unstable readings. This could the attributed to the incomplete polymerization of the monomer species, which in this condition would undergo chemical changes when subject to the electrochemical reading conditions of the probe. The preparation of films within the potential range −0.80 to +0.80 V led to increased currents that revealed the presence of an increased electroactive area, associated with a higher conductivity. In contrast, for the potential range −0.80 to +1.4 V, the electrochemical measurements of the redox probe after successive readings were very stable, but the current signal was much reduced by the formation of a highly insulating film. This result reflected the overoxidation of Py, which was close to +1.1 V [27].

Overall, according to both CV and DPV studies of the different films, the best compromise between electron transfer and sensor surface stability was obtained for PPy films produced within −0.80 to +0.80 V. This experimental condition was used in further studies. 

### 3.3. MWCNT Effect

In general, MWCNTs improve electron transfer and superficial area in electrochemical systems, thereby improving the electrical properties of electrochemical biosensors. Thus, to produce a conductive MPPy of improved electrical features, MWCNTs were added to the reaction mixture, forming a nanocomposite material of PPy/CNT. This nanocomposite is also known to act as a supercapacitor due to its good properties of charge delivery and energy/electron storage. Overall, the inclusion of CNTs within the PPy network formed a 3D structure of increased conductivity, derived from the cross-talk between the conjugated π–π bonds of the PPy structure and of the CNT sidewalls [28]. 

The effect of the addition of MWCNTs to the PPy network was evaluated by CV scanning of the standard redox probe. The data obtained was shown in Figure 4. A reduction of 30% in the peak-to-peak potential separation (ΔE) was observed with the addition of MWCNTs. The typical ΔE of PPy decreased from 0.382 to 0.264 V by adding MWCNTs to the polymerizing mixture, reflecting an improvement in the conductivity of the material. In addition, the faradaic current was also increased by the presence of the MWCNTs, just as expected [29]. 

The effect of the PY was also evaluated by measuring the specific capacitance (Cs). It was calculated by means of Equation (1), in which i is current, V potential window, scan rate and m mass-specific capacitance (F/g). The results indicated that the presence of PPy on the cleaned electrode surface increased the Cs about 12.6%.
Cs = (∫i.dV)/(ν.m.∆V)(1)

### 3.4. Synthesis of the Imprinted Material

The synthesis of the MPPy material was carried out by following the previously selected conditions, adapted to the target compound. Specifically, human Cys-C exists in two isoforms with the isoelectric points 9.2 and 7.8 [30], meaning that it is positively charged in serum, or in pH 6, the pH selected for this work [31]. Thus, the addition of negatively charged monomer species could intensify the binding of Cys-C to the final polymeric network [32]. This would enhance the sensitivity of the final device. 

Thus, the MPPy film was obtained by adding Py-COOH to Cys-C and allowing (over 30 min) self-arrangement between these compounds, by means of ionic interactions. In brief, the negatively charged carboxylic groups of the Py-COOH, at pH 6, were expected to bind to the positively charged amino groups from the protein, by means of ionic interactions. This self-arranged structure was then added to the Py/MWCNT solution, to undergo electropolymerization on the working electrode of the C-SPE. Electropolymerization was conducted by CV, under the selected optimum conditions. Cys-C was then extracted from the polymeric network by treatment with a urea solution (5 µL of 0.1 M of urea dissolved in water were cast on the working electrode area), to leave vacant binding positions with a complementary charge and shape to the target protein. 

The C-SPE surface modification was followed by CV and DPV procedures. The CV profiles of the iron redox probe obtained after formation of the polymeric film (MPPy and NPPy) clearly confirmed the presence of this polymer on the C-SPE (Figure 5). This was confirmed by the increasing of the overall area of the CV voltammograms after formation of the polymeric layer. This behavior also evidenced the presence of a more capacitive surface. 

After the template removal, the overall net current decreased in both the MPPy and NPPy (Figure 5). Although the removal of the Cys-C occurred only in the MPPy, the impact of urea treatment upon electrochemical features of the polymer was more prominent, resulting in insignificant changes for both. This behavior can be attributed to the anionic charge on the Py/PyCOOH that was strongly affected by acid treatment. However, biorecognizing cavities were preserved since only MPPy yielded decreasing net current after the Cys-C incubation. In addition to this, it became clear that after 3 h of incubation with the extraction solution, washing and incubation with the target protein, it was possible to observe a higher binding capacity of the MIP material, when compared with the NIP material. Longer periods of incubation were not tried for technical reasons.

### 3.5. Analytical Performance of the Sensor

The analytical performance of the biosensor was evaluated in acetate buffer, pH 6.0. This was carried out by incubating each standard solution for 20 min, washing it out and reading after the signal of the standard redox probe by squared-wave voltammetry (SWV). The time given for incubation was typically set from 15 to 30 min. In general, longer times may improve sensitivity as there is more time to allow Cys-C adsorption, while shorter times may improve selectivity as there is little time for non-specific adsorption [33,34]. Therefore, as a compromise, without further experiments for this selection, the incubation time was set to an intermediate value of 20 min.

The different Cys-C standard solutions were incubated consecutively in increasing concentrations, up to 30 ng/mL, and the voltammograms obtained are shown in Figure 6 (left). The peaks of the redox probe were centered at 0.15 V and showed decreasing currents (I) for increasing Cys-C concentrations.

The calibrations plotted current (I) responses as a function of the logarithm of Cys-C concentrations, as shown in Figure 6B Under optimum conditions, the MPPy sensor displayed a dynamic response range between 0.5 and 30.0 ng/mL, with a limit of detection (LOD) of 0.5 ng/mL. The typical linear equation was current (mA) = −0.0021 × [Log (Cys-C, ng/mL)] + 0.0024, with a squared correlation coefficient of 0.972 and a standard deviation of repeated assays < 5%.

Identical calibration procedures were performed with the NPPy control films, to assess the dimension of non-specific responses Figure 6C. As these films were prepared without the target protein, there were no binding sites available and any interaction with Cys-C would reveal the presence of a non-specific interaction with the polymer. The results obtained are shown in Figure 6Aand showed a random behavior of the NPPy film in the same range of protein concentration. Thus, this behavior demonstrated that within the concentration range studied, the response of the MPPy was being controlled by the interaction of Cys-C with the imprinted binding sites and a non-specific response was not observed. The reproducibility and repeatability were less than 10%.

### 3.6. Selectivity and Application 

Selectivity was assessed by checking the analytical response of the MPPy devices on a background of diluted spiked Cormay^®^ HN serum, which corresponds to human serum from normal individuals, spiked with specific interfering compounds. The interfering compounds added were creatine kinase-MB 0.2g/L, ascorbic acid 0.15g/L, creatinine 1g/L and bovine serum albumin 12g/L. In general, negligible changes were found when the MIP sensor was incubated with serum samples when compared with blank signal from the buffer. This diluted serum was also spiked with Cys-C to check the ability of the system to respond with accuracy. The electrochemical data were obtained in single sample analysis and the resulting concentration values were extracted from the calibration curve. The spiked samples were prepared in two different Cys-C concentrations, equal to 2.0 and 5.0 ng/mL (Table 1). This analysis was performed in triplicate. The obtained data confirmed a good correlation between added and found amounts of Cys-C, with recovery values ranging from 88 to 99% and standard errors < 5%. This confirmed the accuracy and the reproducibility of the analytical responses. 

## 4. Conclusions

This work produced a selective and stable MIP-based biosensor for the detection of the kidney biomarker Cys-C at PoC. The MPPy was assembled with a quick and simple procedure, yielding good reproducibility, accuracy, and linearity. The use of Py monomers ensured good electrocatalytic properties of the electrode, which were expected to enhance the sensitivity of the electrochemical system. Moreover, the introduction of MWCNTs in the PPy matrix was strategically explored to form a more porous and larger surface, allowing the formation of MPPy 3D structures with good sensitivity. The rebinding ability of this biosensor device was unique when compared to natural biomolecules because it offers high stability and low cost with similar performances. Moreover, this is a disposable device, not tested for regeneration, with suitable features for PoC use.

Overall, this method surpasses previously used methods for monitoring Cys-C in serum, offering faster execution and lower cost. The LODs achieved are of interest for clinical applications, allowing the PoC detection of cystatin C. Moreover, this approach may also be translated to other protein biomarkers.

## Figures and Tables

**Figure 1 biosensors-11-00175-f001:**
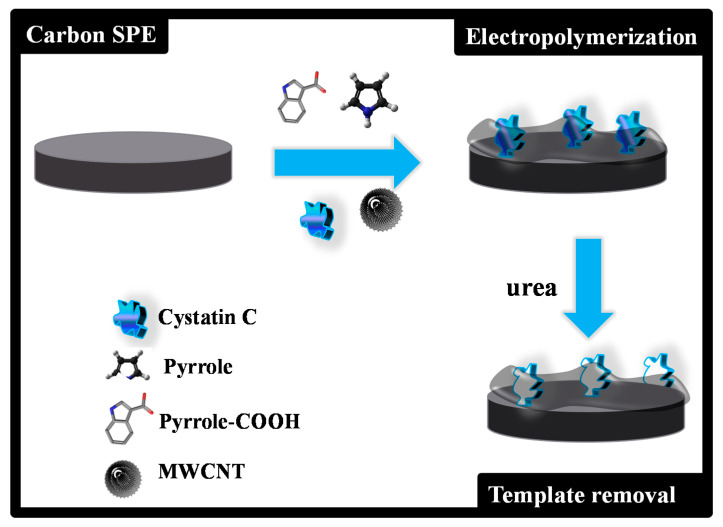
Schematic representation of the assembly process of the imprinted material.

**Figure 2 biosensors-11-00175-f002:**
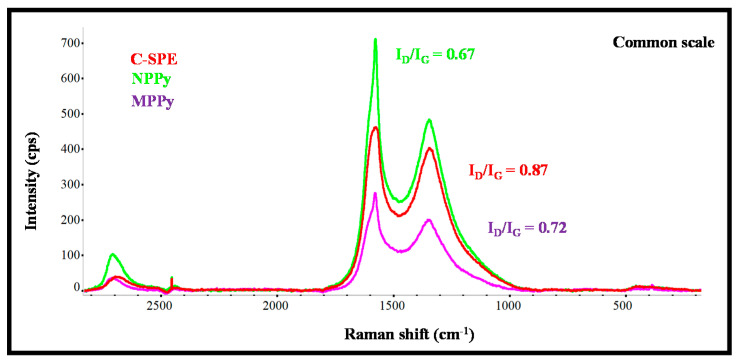
Raman spectra of the carbon working electrode on the SPE (C-SPE) and its subsequent modification with MPPy and NPPy materials.

**Figure 3 biosensors-11-00175-f003:**
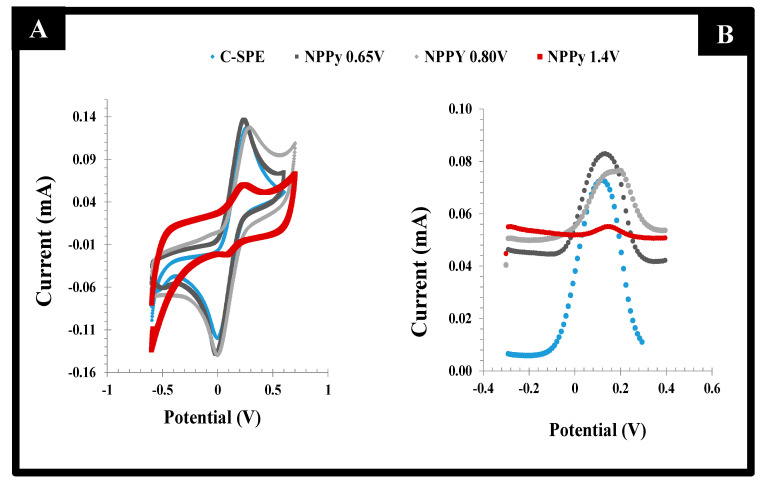
CV (**left**) and DPV (**right**) voltammograms of a 5.0 mmol/L [Fe (CN)6]3−/4− solution in KCl 0.1 mol/L on PPy films prepared with C-SPE electrodes with different potential ranges, starting at −0.80 V and finishing at +0.65, +0.80 or +1.40 V.

**Figure 4 biosensors-11-00175-f004:**
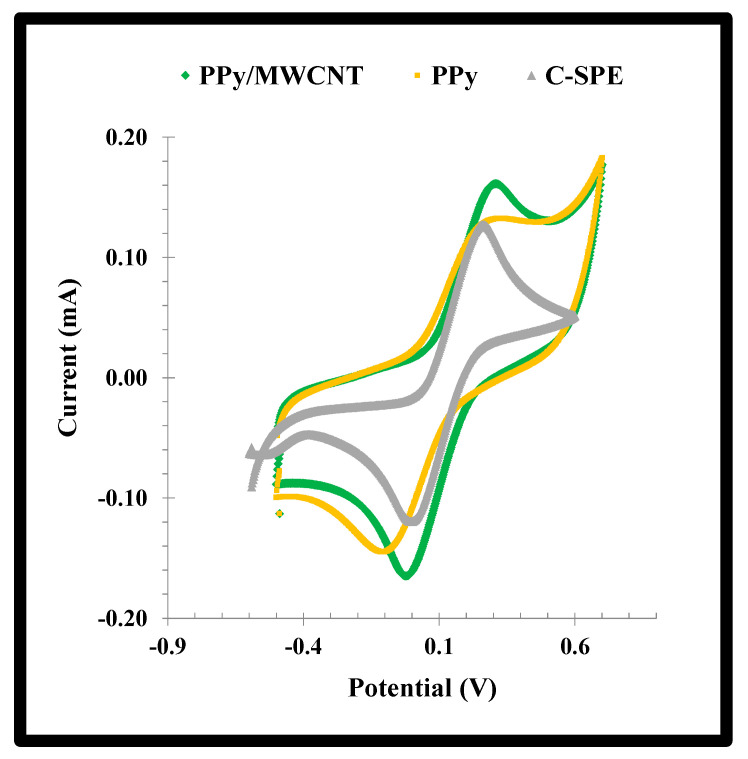
CV voltammograms of a solution of a 5.0 mmol/L [Fe (CN)6]^3−^/^4−^ solution in KCl 0.1 mol/L casted on the NPPy material assembled from −0.80 V to +0.80V, with or without MWCNTs, on substrates of C-SPEs.

**Figure 5 biosensors-11-00175-f005:**
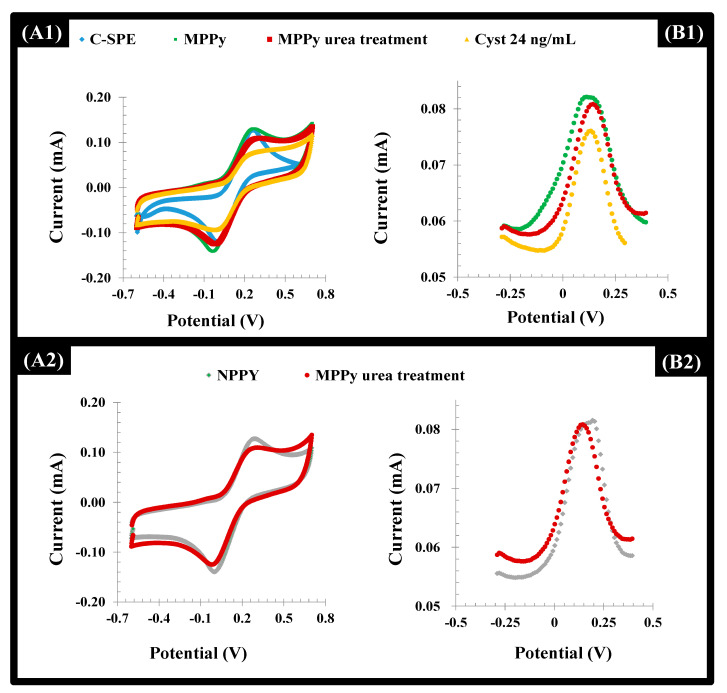
CV (**A**) and DPV (**B**) voltammograms of a 5.0 mmol/L [Fe (CN)6]^3−^/^4−^ solution in KCl 0.1 mol/L corresponding to the assembly of the MPPy and NPPy devices, in the several stages of this process (the blank C-SPE, the electropolymerization on top of it and the subsequent urea treatment), along with the incubation of Cys-C solution (24 ng/mL) on the MPPy film. (**A1** and **A2**)-CV measurements; (**B1** and **B2**):-DPV measurements.

**Figure 6 biosensors-11-00175-f006:**
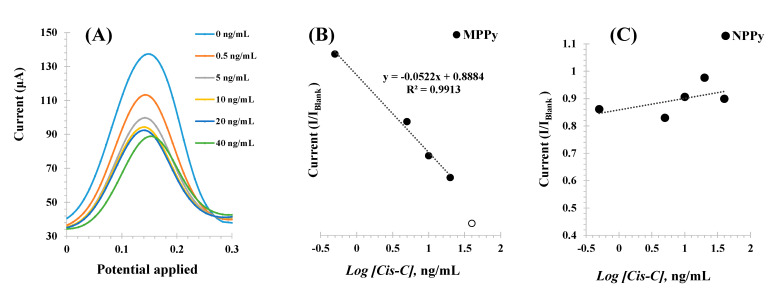
SWV voltammograms (**A**) corresponding to the incubation of increasing concentrations of Cys-C (in ng/mL) and the electrochemical signal obtained with a 5.0 mmol/L [Fe(CN)_6_]^3−/4−^ solution prepared in KCl 0.1 mol/L, and the corresponding calibration curves of the MPPy (**B**) and NPPy (**C**) devices. Assays performed in triplicate.

**Table 1 biosensors-11-00175-t001:** Analytical data obtained with the MPPy biosensor with diluted serum spiked with Cys-C standard solutions.

Sample	Added (ng/mL)	Found (ng/mL)	Recovery (%)	RSD (%)
1	2.0	1.76	87.8	2.20
2	5.0	4.93	98.6	1.42

## Data Availability

Not applicable.

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
