# Peer review of "Plastic Antibody of Polypyrrole/Multiwall Carbon Nanotubes on Screen-Printed Electrodes for Cystatin C Detection"

_biosensors, 2021, doi:10.3390/bios11060175_

Round 1

Reviewer 1 Report

In this work, the authors constructed an electrochemical sensor based on MIP for the detection of the kidney biomarker Cys-C in PoC. The MIP as the specific bonding sites was obtained by electropolymerizing pyrrole (Py) with carboxylated Py (Py-COOH) on the C-SPEs. The binding ability of Cys-C to the MIP material were evaluated by CV and DPV. The prepared biosensor can offer good linearity, reproducibility and low cost. Overall, I think it can be published after major revision. The detailed comments are listed as follows:

(1) The title emphasizes the material of MIP, however, there are very little of material synthesis and characterization in the text. I suggest to include biosensor or detection of Cys-C in the title.

(2) The resolution of Figure 2, especially the x axis and y axis, should be improved so that the readers can clearly understand it.

(3) It is mentioned in this paper that the removement of Cys-C from the polymeric network was realized by the incubation in 0.1M urea for 3h. Here, the urea was used as the eluent of Cys-C, what is the elution efficiency or extraction efficiency? In addition, it is suggested that the surface morphology of the sensor should be characterized by SEM during the modification process, so as to reveal the influence of microstructure change of sensing surface on the performance of this sensor.

(4) The authors think the doping of MWCNTs into the PPy network enables to improve electron transfer and superficial area in electrochemical systems, thus enhancing the sensing performance of this sensor. If CNTs are replaced by gold nanoparticles with better conductivity, can the same effect be achieved? And if the sensing surface is not doped with CNTs, can the Cys-C be detected?  Please make a comment.

(5) In Figure 6, the SWV voltammograms present six curves, in which the current value of the curve marked with 0 ng/mL is about 20 μA higher than that of the curve marked with 0.5 ng/mL, while the difference between the current values of other adjacent curves is less than this value. Does this mean that the sensor can actually achieve a lower detection limit than 0.5 ng/mL? Meanwhile, in the corresponding calibration curve of the MPPy device, why only four points are treated linearly instead of five points?

(6) The scales in Figure 5 (A1) and (A2) needs to be confirmed. Why there are five steps not four steps between the adjacent numbers, i.e. 0.2V and 0.5V (x axis)?

(7) In the selectivity experiments, the authors found that there have no obvious responses of the MIP film when exposed with the diluted Cormay HN serum. Please provide some relevant experimental data either in the main text or in the supplementary to support this conclusion.

(8)  English language and typos need to be improved through the whole manuscript. For example, “in the development a biosensor” should be “in the development of a biosensor”, “The ability binding Cys-C” should be “The binding ability of Cys-C” etc. in the abstract.

Author Response

Manuscript ID: biosensors-1207716

Title: Composite of polypyrrole and multiwall carbon nanotubes to assemble a

plastic antibody for Cystatin C on screen-printed electrodes

Authors: Rui M. Gomes, Blanca Rodriguez, Ruben Fernandes, Goreti Sales, Rosa

Dutra *, Felismina T.C. Moreira *

Please find next  our responses (in black) to the issues raised by reviewers (in blue). All alterations made in the manuscript have been highlighted in yellow.

Comments and Suggestions for Authors

In this work, the authors constructed an electrochemical sensor based on MIP for the detection of the kidney biomarker Cys-C in PoC. The MIP as the specific binding sites was obtained by electropolymerizing pyrrole (Py) with carboxylated Py (Py-COOH) on the C-SPEs. The binding ability of Cys-C to the MIP material was evaluated by CV and DPV. The prepared biosensor can offer good linearity, reproducibility, and low cost. Overall, I think it can be published after major revision. The detailed comments are listed as follows:

  • The title emphasizes the material of MIP, however, there are very little of material synthesis and characterization in the text. I suggest to include biosensor or detection of Cys-C in the title.

We agree with the reviewer and the title was modified as suggested. The title now goes as “Plastic antibody of polypyrrol/multiwall carbon nanotubes on screen-printed electrodes for an innovative Cystatin C detection”. Please check this in the revised paper.

  • The resolution of Figure 2, especially the x axis and y axis, should be improved so that the readers can clearly understand it.

The resolution of the image has been improved. Please see the new figure 2, in p. 6.

  • It is mentioned in this paper that the removement of Cys-C from the polymeric network was realized by the incubation in 0.1M urea for 3h. Here, the urea was used as the eluent of Cys-C, what is the elution efficiency or extraction efficiency?

Overall, in electrochemical sensing this is done by following the electrochemical response. In brief, the efficiency of template removal was evaluated by comparing the net-current (after and before) of the template removal and by comparing the (re)binding capacity of the MIP and NIP-based sensors.  Overall, we found the best operational features for the MIP-based sensors were achieved when the protein was removed under these conditions (incubation in 0.1M urea for 3h)

  1. In addition, it is suggested that the surface morphology of the sensor should be characterized by SEM during the modification process, so as to reveal the influence of microstructure change of sensing surface on the performance of this sensor.

We consider this a good suggestion, however, as the electrode is printed with a very rough and thick layer of carbon ink, this makes it difficult to clearly distinguish the conductive polymeric material deposited on its surface, which has a thickness lying within the nm range. We have tried this in previous works involving plastic antibodies prepared by electropolymerization and the results were not satisfactory in terms of sensitivity to detect the really important nanoscale modifications played by the imprinted proteins.

  1. The authors think the doping of MWCNTs into the PPy network enables to improve electron transfer and superficial area in electrochemical systems, thus enhancing the sensing performance of this sensor. If CNTs are replaced by gold nanoparticles with better conductivity, can the same effect be achieved? And if the sensing surface is not doped with CNTs, can the Cys-C be detected?  Please make a comment.

Other types of highly conductive nanomaterials as graphene, gold or platinum nanoparticles could also be used for the same purpose, as all these show exceptional features in terms of surface area and conductivity. In this work, we selected the carbon nanotubes due to their outstanding features in terms of electrical and superficial area. The long rod shape of MWCNTs also enhances the interconnection between the conductive materials on the polymeric network, which increases more the overall conductivity properties of the electrode, when compared to the typical spherical shape of metal nanoparticles. In addition, carbon-based materials are less expensive and more sustainable than metal ones.

In terms of analytical output, it is expected that the sensitivity of the response decreases, because the presence of the MWCNTs increased the net current, as shown in Figure 4. But this does not mean that Cys-C would not be detected, considering that there is a wide current difference still between the blank and the first standard solution used (Figure 6). So, we would expect that Cys-C would be detected but with lower sensitivity.

  1. In Figure 6, the SWV voltammograms present six curves, in which the current value of the curve marked with 0 ng/mL is about 20 μA higher than that of the curve marked with 0.5 ng/mL, while the difference between the current values of other adjacent curves is less than this value. Does this mean that the sensor can actually achieve a lower detection limit than 0.5 ng/mL?

Yes, we agree with the reviewer. It is likely possible to obtain an extended linear range for lower concentrations values than 0.5 ng/mL, although for clinical interest the selected concentration range is adequate. This also justifies stating that the limit of detection would be lower than 0.5 ng/mL.

  1. Meanwhile, in the corresponding calibration curve of the MPPy device, why only four points are treated linearly instead of five points?

The reviewer is right. Was a mistake in the figure on the left. The analytical response of the electrode is linear within 0.5 to 20.0 ng/mL. The last point of the graph is related to the standard 30 ng/mL. This value of concentration was out of the linear range and consequently was not considered in the linear range of the calibration curve. If you carefully observe, the point is present in the figure but in a different format (circle not fulfilled). This was updated in the manuscript.

  1. The scales in Figure 5 (A1) and (A2) needs to be confirmed. Why there are five steps not four steps between the adjacent numbers, i.e. 0.2V and 0.5V (x axis)?

The figure has been updated as the reviewer suggested. In addition, the yy axis of Figures 3 to 5 was also corrected, as V was used by mistake instead of A.

  1. In the selectivity experiments, the authors found that there have no obvious responses of the MIP film when exposed to the diluted Cormay HN serum. Please provide some relevant experimental data either in the main text or in the supplementary to support this conclusion.

The reviewer is right.  Some information is missing.

The sentence “Negligible changes were found, thereby confirming the selective response of the MIP film when in presence of a very complex sample, which is indeed the final conditions in which the biosensor shall be tested. “ is changed and replaced by:

Negligible changes were found when the MIP sensor was incubated with serum samples when compared with the blanc signal from de buffer, thereby confirming the selective response of the MIP film when in presence of a very complex sample, which is indeed the final conditions in which the biosensor shall be tested.

Overall, when the MIP sensor was incubated with serum samples, the blank signal changed less than 10 % to that of the buffer.

  1. English language and typos need to be improved throughout the whole manuscript. For example, “in the development a biosensor” should be “in the development of a biosensor”, “The ability binding Cys-C” should be “The binding ability of Cys-C” etc. in the abstract.

We have revised the manuscript throughout, as suggested. These changes were not highlighted.

Reviewer 2 Report

The manuscript with the title Composite of polypyrrol and multiwall carbon nanotubes to assemble a plastic antibody for Cystatin C on screen-printed electrodes presents the development of a MIP modified SPE for Cystatin-C detection. The topic is interesting and worth studying, but, in my opinion, the whole procedure and the results are not good explained in the manuscript.

Below are listed some suggestions that hopefully may help to improve the manuscript in order to bring it in a form that could be accepted for publication.

  1. I did not understand why the C-SPE must be modified, because the authors did not show the advantages of the electrode modification in comparison to the bare C-SPE. Why is it not enough to measure the decrease of the ferricyanide/ferrocyanide signal at the bare C-SPE decrease by adding in the solution Cys-C?
  2. The abstract must be more concise and more exact.
  3. Abstract: "Casting increasing concentrations of Cys-C (0.5 to 30.0 ng/mL) on the MPPy surface for 20 min, yielded increased redox currents of the same ferro/ferricyanide standard solution." -

This statement is not in accordance with the presented results because the authors detected Cys-C based on the decrease of the probe signal in the presence of Cys-C.

  1. Introduction: The abbreviation PoC should be explained when it appears for the first time in the text.
  2. Introduction: "based on" instead of "base on"
  3. Beside the used instrument the authors should indicate also the program used for the voltammetric data acquisition.
  4. The provenience of MWCNTs should be also indicated in section 2.2. Reagents and solutions.
  5. "Selectivity studies used synthetic serum spiked with other compounds that could act as interfering species (Creatine kinase-MB (CK-MB) 0.2g/L, AA 0.15g/L, Crea 1g/L and BSA 12g/L)." -

This statement from section 2.2 was not confirmed by presenting the results and discussions in section 3.6 Selectivity and Application.

  1. "Electrochemical readings were obtained for MPPy and NPPy materials (minimum, n=3)." Which is the significance of "minimum, n=3"?
  2. "The MPPy material was obtained in a pH 6 acetate buffer solution containing MWCNT (40 %), Py (80.0 mol/L), 10% Py-COOH (4.0 mmol/L) and 1% Cys-C (0.050 µg/ml)." - How was this composition of the electropolymerizing solution selected?
  3. "The extraction of Cys-C from the polymeric network was made by incubating on the working electrode area of the MPPy/C-SPEs in 0.1M urea, for 3 hours" - How was the incubation time selected? Why 3 hours?
  4. The compositions of the solutions used for obtaining the NPPy and of the PPy/MWCNT films by electropolymerization on C-SPE should be clearly indicated.
  5. According to Fig. 3 the highest DPV signal was obtained on the unmodified C-SPE and the CVs on NPPy modified C-SPE are not very different from those obtained at bare C-SPE. In this conditions, the authors should better explain why did they select to use further the electrodes modified with NPPy obtained by electropolymerization in the range -0.8 to 0.8 V.
  6. The authors showed that they studied the influence of the cycling potential on the NPPy formation, but they didn’t wrote anything about how / why they selected the number of cycles (10) and the scan rate (20 mV/s)?
  7. "Overall, the inclusion of CNTs within the PPy network formed a 3D structure of increased conductivity, derived from the cross-talk between the conjugated π-π bonds of the PPy structure and of the CNT sidewalls [27]." - Maybe some surface characterization techniques (e.g. SEM) should be used to support this statement and to show the differences between the different surfaces (C, NPPY, MPPY before and after urea treatment).
  8. According to figure 4 the improvement on the CVs due to C-SPE modification is not clear. On bare C-SPE the signals are better defined (with lower nackground current, almost the same peak height as on the PPy/MWCNT/C-SPE but sharper) than on modified electrodes and the peak-to-peak separation is almost the same at C-SPE and PPy/MWCNT/C-SPE.
  9. "the MPPy film was obtained by addting Py-COOH to Cys-C and allowing (in time) the formation of a complex between these compounds, by means of ionic-interactions" - What means "in time"? The exact reaction time should be given.
  10. How was Py-COOH added to Cys-C? I suppose that solutions of the two compounds were mixed. Exact experimental details should be given.
  11. "This complex was then added to the Py/MWCNT solution," How was this complex (between Py-COOH and Cys-C) emphasized? Was the complex extracted from the reaction mixture? Was it purified and dissolved in a solvent or was the solution containing the reaction mixture simply added to the Py/MWCNT solution? The procedure of "adding" the "complex" to the polymerization solution should be exactly given.
  12. "Cys-C was then extracted from the polymeric network by treatment with a urea solution" - Exact experimental details should be given (urea concentration, solvent used for the urea solution, extraction time).
  13. In my opinion the indication "of the MPPy (1) and NPPy (2) devices" in the legend of Figure 5 is not correct and it should be deleted because in all 4 images (A1, B1, A2 and B2) are shown different comparative overlaid voltammograms.
  14. For me it is not clear what is exactly represented in Figura 5A and 5B1. I understand that Figure 5A1 represents CVs of the [Fe (CN)6]3-/4- redox couple at C-SPE (blue line), MPPy at C-SPE (green) MPPY after urea treatment (red). I do not understand what represents the yellow CV? What represents the legend “24 ng/mL Cyst” for yellow? The same is valid for Figure 5B1 but here I do not see the DPV at C-SPE which must be given as comparison. If the yellow voltammograms correspond to the CVs/DPV of the [Fe (CN)6]3-/4- redox couple in the presence of Cys-C then it must be indicated at which of the discussed electrodes were they obtained.
  15. According to figure 5 I do not see an improvement of the redox probe signal at the modified electrodes in comparison to the bare C-SPE.
  16. "After template removal the overall net current decreased (Figure 5). As this decrease was observed in both MPPy and NPPy, this behaviour revealed not only the removal of Cys-C but mostly the impact of the urea treatment upon the electrochemical features of the polymer. This is because the NPPy had no Cys-C to remove and therefore should not change. This effect was more intense in the MPPy, meaning that this was an outcome of the presence of Cys-C on the substrate. In addition, the NPPy polymer presented higher peak current when compared with the MPPy after template removal. This could be attributed to the presence of Cys-C in the polymerization stage of MPPy polymer." - According to this statement, the current decreases after template removal and this effect was observed at both NPPy and MPPy. Which is the relation between NPPy and the template removal and the ureea treatment? The whole explanation is not clear for me.
  17. "This was made by incubating for 20 min each standard solution". - How was the incubation time selected? The sentence should be more exact. The standard solution of what compound?
  18. "with a limit of detection (LOD) <0.5 ng/mL" - An exact value of LOD should be given.
  19. Was the used electrode a disposable one or was it used several times? If it was not disposable than aspects like: stability and electrode regeneration/ cleaning procedure should be discussed.
  20. What about the reproducibility and repeatability of the electrochemical response?

Author Response

The manuscript with the title Composite of polypyrrole and multiwall carbon nanotubes to assemble a plastic antibody for Cystatin C on screen-printed electrodes presents the development of a MIP modified SPE for Cystatin-C detection. The topic is interesting and worth studying, but, in my opinion, the whole procedure and the results are not well explained in the manuscript.

Below are listed some suggestions that hopefully may help to improve the manuscript in order to bring it in a form that could be accepted for publication.

  1. I did not understand why the C-SPE must be modified, because the authors did not show the advantages of the electrode modification in comparison to the bare C-SPE. Why is it not enough to measure the decrease of the ferricyanide/ferrocyanide signal at the bare C-SPE decrease by adding in the solution Cys-C?

A molecularly imprinted polymer (MIP), also known as a plastic antibody, works as a biorecognition layer that ensures that the electrochemical response obtained comes from Cys-C. Without this MIP layer, a response in real samples would be linked to all compounds present in the sample and not to only Cys-C, as the carbon material would absorb/adsorb any protein/biomolecule present. Yet, this work targets monitoring Cys-C in particular, and therefore a biorecognition layer becomes fundamental.

Moreover, MIPs are considered biomimetic materials with similar characteristics to their natural counterparts, the antibodies obtained from biological sources. These polymers can selectively recognize a given target molecule to which they were designed. If applied as recognition units of biosensors, these receptors provide very high selectivity, justifying that the use of MIPs as recognition units in biochemical sensors is gaining increasing interest.

  1. The abstract must be more concise and more exact.

The abstract has been condensed, trying to respond to your request. Please see the new abstract in p. 2.

  1. Abstract: "Casting increasing concentrations of Cys-C (0.5 to 30.0 ng/mL) on the MPPy surface for 20 min, yielded increased redox currents of the same ferro/ferricyanide standard solution." –This statement is not in accordance with the presented results because the authors detected Cys-C based on the decrease of the probe signal in the presence of Cys-C.

This was a spelling error. With the increase of protein concentration, we observed a decrease in the net current. This is now correct (please see p. 2).

  1. Introduction: The abbreviation PoC should be explained when it appears for the first time in the text.

Sorry for this. The abbreviation was described in the manuscript (please see p. 3).

  1. Introduction: "based on" instead of "base on"

The sentence has been changed.

  1. Besides the used instrument the authors should indicate also the program used for the voltammetric data acquisition.

The program was NOVA software (the most recent version). Please see this information in p. 4.

  1. The provenience of MWCNTs should be also indicated in section 2.2. Reagents and solutions.

The nanotubes were obtained from Fluka. This information is now shown in p. 4.

  1. "Selectivity studies used synthetic serum spiked with other compounds that could act as interfering species (Creatine kinase-MB (CK-MB) 0.2g/L, AA 0.15g/L, Crea 1g/L, and BSA 12g/L)." -This statement from section 2.2 was not confirmed by presenting the results and discussions in section 3.6 Selectivity and Application.

The results presented in the table in section 3.6 represent the selectivity study. The serum was dopped with those compounds, and then, spiked with different concentrations of Cys-C. Overall, the results demonstrated that it was possible to distinguish selectively the target compound in a very complex matrix.

  1. "Electrochemical readings were obtained for MPPy and NPPy materials (minimum, n=3)." Which is the significance of "minimum, n=3"?

We agree this is important information. This means that at least three readings were considered in the data presented herein. The information was changed to “a minimum of three replicate readings (n<3).” We hope this clarifies.

  1. "The MPPy material was obtained in a pH 6 acetate buffer solution containing MWCNT (40 %), Py (80.0 mol/L), 10% Py-COOH (4.0 mmol/L) and 1% Cys-C (0.050 µg/ml)." - How was this composition of the electropolymerizing solution selected?

This composition was selected according to the group experience and optimization steps. First, the use of a small amount of Py-COOH compared to Py follows the same principle as that of the preparation of a material we signaled as SPAM (https://doi.org/10.1016/j.bios.2013.02.012). In this, the protein is surrounded by a monomer that is different from the overall polymeric matrix, aiming to enhance the capacity of recognizing this protein (it shall hold a higher affinity to the binding site). The use of pH 6 and 1% Cys-C is typically linked to previous preliminary experiments involving other protein imprinting assemblies.

  1. "The extraction of Cys-C from the polymeric network was made by incubating on the working electrode area of the MPPy/C-SPEs in 0.1M urea, for 3 hours" - How was the incubation time selected? Why 3 hours?

Our experience lets us know that few hours are required to achieve suitable data but a long time, as 10h (overnight) may disturb the sensing layer. In addition, as the assembly of the electrodes is done in one day and the calibration in the following day, more than 3 hours and less than overnight (10h) become technically impossible. Thus, our studies are typically ranging removal timings up to 3h, depending on the polymer, the protein, and the other (electro)chemical conditions. In this range, longer times as 3h should be better, provided that the polymer does not change its electrochemical behavior in an intense manner that may suggest enhanced chemical alterations.

Herein, after 3h of incubation with the extraction solution, washing, and incubation with the target protein, it was possible to observe a higher binding capacity of the MIP material, when compared with the NIP material. We found this an acceptable result and proceed with it.

  1. According to Fig. 3 the highest DPV signal was obtained on the unmodified C-SPE and the CVs on NPPy modified C-SPE are not very different from those obtained at bare C-SPE. In these conditions, the authors should better explain why did they select to use further the electrodes modified with NPPy obtained by electropolymerization in the range -0.8 to 0.8 V.

As explained previously, the use of a biorecognition layer is fundamental and most of the time it imposes a significant reduction upon the electrical signal of the SPE. Not happening here is already an important result. Moreover, the CV and DPV spectra of C-SPE and NIP material are significantly different in terms of capacitance. The CV and DPV of NIP material show higher area and capacitive currents. The spectra of C-SPE are representative of the faradaic process.

  1. In this condition, the authors should better explain why did they select to use further the electrodes modified with NPPy obtained by electropolymerization in the range -0.8 to 0.8 V.

In general, the increasing potentials increased the extent of polymer formation, as more energy was being introduced into the electrochemical system. While the higher potential values had the possibility of producing the over-oxidation of the film, the lower values displayed poorer reproducibility. This poor reproducibility reflected the fact that the voltage required to reach the maximum Py oxidation (maximum current) was far from being achieved. Thus, the selected range of potential was within -0.80 to +0.80V, which ensured the production of a reproducible and stable polymeric layer.

This information in now on the manuscript. Please see p. 7.

  1. The authors showed that they studied the influence of the cycling potential on the NPPy formation, but they didn’t wrote anything about how / why they selected the number of cycles (10) and the scan rate (20 mV/s)?

These studies were based on the stability of the sensor after successive readings with redox probe and washings steps with the buffer. Within these conditions, the sensor showed stable electrochemical measurements.  

  1. "Overall, the inclusion of CNTs within the PPy network formed a 3D structure of increased conductivity, derived from the cross-talk between the conjugated π-π bonds of the PPy structure and of the CNT sidewalls [27]." - Maybe some surface characterization techniques (e.g. SEM) should be used to support this statement and to show the differences between the different surfaces (C, NPPY, MPPY before and after urea treatment).

We consider this a good suggestion, but SEM analysis will not allow us to confirm these π-π bonds and the increased conductivity is confirmed with data in Figure 3. Moreover, from an organic chemistry point of view, it is only logical that this is taking place when aromatic organic structures can interact, which is the case of this study (also confirmed in the citation).

  1. According to figure 4 the improvement on the CVs due to C-SPE modification is not clear. On bare C-SPE the signals are better defined (with lower background current, almost the same peak height as on the PPy/MWCNT/C-SPE but sharper) than on modified electrodes and the peak-to-peak separation is almost the same at C-SPE and PPy/MWCNT/C-SPE.

The main and relevant difference between the C-SPE and the modified electrodes are the capacitive currents observed, due to the presence of a capacitor on the electrode surface. In this case, the capacitor is the PPy. When we added MWCNT, it was possible to observe an increment of the overall current values, which is expected once the MWCNT is a very conductive nanomaterial. Moreover, the current levels at about 0.7V are much greater in PPy/MWCNT and PPy than in C-SPE.

  1. "the MPPy film was obtained by adding Py-COOH to Cys-C and allowing (in time) the formation of a complex between these compounds, by means of ionic-interactions" - What means "in time"? The exact reaction time should be given.

The protein was let react with the PPy for 30 minutes. This information is now in p. 10.

  1. How was Py-COOH added to Cys-C? I suppose that solutions of the two compounds were mixed. Exact experimental details should be given.

Yes, these compounds were mixed to match the overall composition already indicated in section 2.4. MPPy material was obtained in a pH 6 acetate buffer solution containing MWCNT (40 %), Py (80.0 mol/L), 10% Py-COOH (4.0 mmol/L) and 1% Cys-C (0.050 µg/ml).

  1. "This complex was then added to the Py/MWCNT solution," How was this complex (between Py-COOH and Cys-C) emphasized? Was the complex extracted from the reaction mixture? Was it purified and dissolved in a solvent or was the solution containing the reaction mixture simply added to the Py/MWCNT solution? The procedure of "adding" the "complex" to the polymerization solution should be exactly given.

In a trivial deduction, the negatively charged carboxylic groups of the Py-COOH, at pH 6, binds to the positively charged amino groups from the protein by means of ionic interactions. Of course, further details cannot be given herein, as this involves having data about the 3D structure/spatial arrangement of Cys-C under these specific experimental conditions, which goes far beyond this work and lies outside our expertise.

From our understanding, the reviewer’s concerns are related to the word “complex”. This terminology is typical in MIPs assembly, but we have replaced the word “complex” by “self-organized arrangement”. Please see this in p. 10.

  1. "Cys-C was then extracted from the polymeric network by treatment with a urea solution" - Exact experimental details should be given (urea concentration, solvent used for the urea solution, extraction time).

For this purpose, a drop of 5 µL of 0.1M of urea dissolved in water was cast on the electrode surface. This information is now in the manuscript.

  1. In my opinion, the indication "of the MPPy (1) and NPPy (2) devices" in the legend of Figure 5 is not correct and it should be deleted because in all 4 images (A1, B1, A2 and B2) are shown different comparative overlaid voltammograms.

The reviewer is right. This was a mistake and has been corrected.

  1. For me, it is not clear what is exactly represented in Figura 5A and 5B1. I understand that Figure 5A1 represents CVs of the [Fe (CN)6]3-/4- redox couple at C-SPE (blue line), MPPy at C-SPE (green) MPPY after urea treatment (red). I do not understand what represents the yellow CV? What represents the legend “24 ng/mL Cyst” for yellow? The same is valid for Figure 5B1 but here I do not see the DPV at C-SPE which must be given as a comparison. If the yellow voltammograms correspond to the CVs/DPV of the [Fe (CN)6]3-/4- redox couple in the presence of Cys-C then it must be indicated at which of the discussed electrodes were they obtained.

The yellow CV represents the binding assay when the MIP-based sensor was incubated with the target protein 24 ng/mL, as indicated in the legend.  After template binding, the electrode was washed, and the electrochemical readings were obtained against the iron redox probe.

  1. According to figure 5, I do not see an improvement of the redox probe signal at the modified electrodes in comparison to the bare C-SPE.

We understand your concerns but as explained before we are not trying to improve the C-SPE performance but instead to assemble an efficient biorecognition layer on top of the working electrode that promotes a small electrochemical impact upon the overall C-SPE response.

  1. "After template removal, the overall net current decreased (Figure 5). As this decrease was observed in both MPPy and NPPy, this behavior revealed not only the removal of Cys-C but mostly the impact of the urea treatment upon the electrochemical features of the polymer. This is because the NPPy had no Cys-C to remove and therefore should not change. This effect was more intense in the MPPy, meaning that this was an outcome of the presence of Cys-C on the substrate. In addition, the NPPy polymer presented a higher peak current when compared with the MPPy after template removal. This could be attributed to the presence of Cys-C in the polymerization stage of MPPy polymer." - According to this statement, the current decreases after template removal, and this effect was observed at both NPPy and MPPy. Which is the relation between NPPy and the template removal and the urea treatment? The whole explanation is not clear to me.

We agree with the reviewer. Text was clarified as this:

After the template removal, the overall net current decreased in both the MPPy and NPPy (Figure 5). Although the removal of the Cys-C has occurred only in MPPy, the impact of urea treatment upon electrochemical features of the polymer was more prominent resulting in significant changes for both. This behavior can be attributed to the anionic charge on the Py/PyCOOH that was strongly affected by acid treatment. However, recognizing cavities were preserved since only in the MPPy was observed a decreasing on net current after the Cys-C incubating.  

Please see p. 11.

  1. "This was made by incubating for 20 min each standard solution". - How was the incubation time selected? The sentence should be more exact. The standard solution of what compound?

This time was selected according to the previous studies of the group working in the field of electrochemical biosensors based on molecular imprinting.

Typically, incubations times are set to 15-30 minutes. Longer times may improve sensitivity (longer time to allow Cys-C adsorption), shorter times improve selectivity (little time for non-specific adsorption). Therefore, as a compromise of this experience, without further experiments for this selection, the incubation time was set herein to an intermediate value of 20 min.

  1. "with a limit of detection (LOD) <0.5 ng/mL" - An exact value of LOD should be given.

This means that the LOD could not be estimated, because the calibration curve is set in log scale of concentration.  As it must below the linear response range, we decided to say it was below 0.5 ng/mL.

  1. Was the used electrode a disposable one or was it used several times? If it was not disposable than aspects like: stability and electrode regeneration/ cleaning procedure should be discussed.

The electrode was not reused or regenerated (this information is now in p. 14).

  1. What about the reproducibility and repeatability of the electrochemical response?

The reproducibility and repeatability were less than 10%.

Round 2

Reviewer 1 Report

The revised manuscript has been improved and all of the questions are addressed. There is still one problem in the current version: The units in figure 5 B1 and B2 still need to be corrected - they are mA, not mV. 

It can be published after the next revision.

Author Response

Reviewer 1

The revised manuscript has been improved and all of the questions are addressed. There is still one problem in the current version: The units in figure 5 B1 and B2 still need to be corrected - they are mA, not mV.  It can be published after the next revision.

Response:  We agree with the reviewer. The figure was updated. Please see p. 13.

Reviewer 2 Report

 I have carefully read the revised form of the manuscript along with the authors’ responses to the raised issues.

Mostly I agree with the answers given but I believe that some information should be included in the manuscript. My suggestions are listed below:

1. I agree with the authors’ clear response to my question: "I did not understand why the C-SPE must be modified, because the authors did not show the advantages of the electrode modification in comparison to the bare C-SPE. Why is it not enough to measure the decrease of the ferricyanide/ferrocyanide signal at the bare C-SPE decrease by adding in the solution Cys-C?"

In order to better emphasize the novelty and importance of the study I recommend to introduce this explanation in the Introduction: "A molecularly imprinted polymer (MIP), also known as a plastic antibody, works as a biorecognition layer that ensures that the electrochemical response obtained comes from Cys-C. Without this MIP layer, a response in real samples would be linked to all compounds present in the sample and not to only Cys-C, as the carbon material would absorb/adsorb any protein/biomolecule present. Yet, this work targets monitoring Cys-C in particular, and therefore a biorecognition layer becomes fundamental.

Moreover, MIPs are considered biomimetic materials with similar characteristics to their natural counterparts, the antibodies obtained from biological sources. These polymers can selectively recognize a given target molecule to which they were designed. If applied as recognition units of biosensors, these receptors provide very high selectivity, justifying that the use of MIPs as recognition units in biochemical sensors is gaining increasing interest."

2. I totally agree with the authors’ response: "This composition was selected according to the group experience and optimization steps. First, the use of a small amount of Py-COOH compared to Py follows the same principle as that of the preparation of a material we signaled as SPAM (https://doi.org/10.1016/j.bios.2013.02.012). In this, the protein is surrounded by a monomer that is different from the overall polymeric matrix, aiming to enhance the capacity of recognizing this protein (it shall hold a higher affinity to the binding site). The use of pH 6 and 1% Cys-C is typically linked to previous preliminary experiments involving other protein imprinting assemblies." to the following issue: "The MPPy material was obtained in a pH 6 acetate buffer solution containing MWCNT (40 %), Py (80.0 mol/L), 10% Py-COOH (4.0 mmol/L) and 1% Cys-C (0.050 µg/ml)." - How was this composition of the electropolymerizing solution selected?"

but I recommend to specify this briefly in the manuscript or to give a reference.

3. I totally agree with the authors’ response: "Our experience lets us know that few hours are required to achieve suitable data but a long time, as 10h (overnight) may disturb the sensing layer. In addition, as the assembly of the electrodes is done in one day and the calibration in the following day, more than 3 hours and less than overnight (10h) become technically impossible. Thus, our studies are typically ranging removal timings up to 3h, depending on the polymer, the protein, and the other (electro)chemical conditions. In this range, longer times as 3h should be better, provided that the polymer does not change its electrochemical behavior in an intense manner that may suggest enhanced chemical alterations.

Herein, after 3h of incubation with the extraction solution, washing, and incubation with the target protein, it was possible to observe a higher binding capacity of the MIP material, when compared with the NIP material. We found this an acceptable result and proceed with it." to the following issue: "The extraction of Cys-C from the polymeric network was made by incubating on the working electrode area of the MPPy/C-SPEs in 0.1M urea, for 3 hours" - How was the incubation time selected? Why 3 hours?"

but I recommend to specify this briefly in the manuscript or to give a reference.

 4. I totally agree with the authors’ response: "This time was selected according to the previous studies of the group working in the field of electrochemical biosensors based on molecular imprinting.

Typically, incubations times are set to 15-30 minutes. Longer times may improve sensitivity (longer time to allow Cys-C adsorption), shorter times improve selectivity (little time for non-specific adsorption). Therefore, as a compromise of this experience, without further experiments for this selection, the incubation time was set herein to an intermediate value of 20 min." to the following issue: "This was made by incubating for 20 min each standard solution". - How was the incubation time selected? The sentence should be more exact. The standard solution of what compound?"

but I recommend to specify this briefly in the manuscript or to give a reference.

5. I totally agree with the authors’ response: "These studies were based on the stability of the sensor after successive readings with redox probe and washings steps with the buffer. Within these conditions, the sensor showed stable electrochemical measurements." to the following issue: "The authors showed that they studied the influence of the cycling potential on the NPPy formation, but they didn’t wrote anything about how / why they selected the number of cycles (10) and the scan rate (20 mV/s)?"

but I recommend to specify this briefly in the manuscript

6. Why did the authors change the OX axis of the calibration curves for MPPy and NPPY from log [Cyst, c] to Potential Applied (Figure 6)?????? Please verify and correct!

 7. The authors have specified now in the text that the used human serum from normal individuals was "spiked with specific interfering compounds" but I recommend to list here again the added interfering species (Creatine kinase-MB (CK-MB) 0.2g/L, AA 0.15g/L, Crea 1g/L and BSA 12g/L) which were listed in section 2.2.

Despite the fact that the abbreviations are obvious I recommend to explain Cre and BSA.

Author Response

 I have carefully read the revised form of the manuscript along with the authors’ responses to the raised issues.

Mostly I agree with the answers given but I believe that some information should be included in the manuscript. My suggestions are listed below:

  1. I agree with the authors’ clear response to my question: "I did not understand why the C-SPE must be modified, because the authors did not show the advantages of the electrode modification in comparison to the bare C-SPE. Why is it not enough to measure the decrease of the ferricyanide/ferrocyanide signal at the bare C-SPE decrease by adding in the solution Cys-C?" In order to better emphasize the novelty and importance of the study I recommend to introduce this explanation in the Introduction: "

Response: We acknowledge your comment. We added this information to the manuscript.

Please see pp. 3 and 4.

  1. I totally agree with the authors’ response: "This composition was selected according to the group experience and optimization steps. First, the use of a small amount of Py-COOH compared to Py follows the same principle as that of the preparation of a material we signaled as SPAM (https://doi.org/10.1016/j.bios.2013.02.012). In this, the protein is surrounded by a monomer that is different from the overall polymeric matrix, aiming to enhance the capacity of recognizing this protein (it shall hold a higher affinity to the binding site). The use of pH 6 and 1% Cys-C is typically linked to previous preliminary experiments involving other protein imprinting assemblies." to the following issue: "The MPPy material was obtained in a pH 6 acetate buffer solution containing MWCNT (40 %), Py (80.0 mol/L), 10% Py-COOH (4.0 mmol/L) and 1% Cys-C (0.050 µg/ml)." – How was this composition of the electropolymerizing solution selected?" but I recommend to specify this briefly in the manuscript or to give a reference.

Response: We acknowledge your comment. We added this information to the manuscript, along with supporting references.

Please see p. 6 and the additional reference in p. 19.

  1. I totally agree with the authors’ response: "Our experience lets us know that few hours are required to achieve suitable data but a long time, as 10h (overnight) may disturb the sensing layer. In addition, as the assembly of the electrodes is done in one day and the calibration in the following day, more than 3 hours and less than overnight (10h) become technically impossible. Thus, our studies are typically ranging removal timings up to 3h, depending on the polymer, the protein, and the other (electro)chemical conditions. In this range, longer times as 3h should be better, provided that the polymer does not change its electrochemical behavior in an intense manner that may suggest enhanced chemical alterations.

Herein, after 3h of incubation with the extraction solution, washing, and incubation with the target protein, it was possible to observe a higher binding capacity of the MIP material, when compared with the NIP material. We found this an acceptable result and proceed with it." to the following issue: "The extraction of Cys-C from the polymeric network was made by incubating on the working electrode area of the MPPy/C-SPEs in 0.1M urea, for 3 hours" - How was the incubation time selected? Why 3 hours?" but I recommend to specify this briefly in the manuscript or to give a reference.

Response: This information was specified in the manuscript.  Please check pp. 13.

  1. I totally agree with the authors’ response: "This time was selected according to the previous studies of the group working in the field of electrochemical biosensors based on molecular imprinting.

Typically, incubations times are set to 15-30 minutes. Longer times may improve sensitivity (longer time to allow Cys-C adsorption), shorter times improve selectivity (little time for non-specific adsorption). Therefore, as a compromise of this experience, without further experiments for this selection, the incubation time was set herein to an intermediate value of 20 min." to the following issue: "This was made by incubating for 20 min each standard solution". –

How was the incubation time selected? The sentence should be more exact. The standard solution of what compound?"but I recommend to specify this briefly in the manuscript or to give a reference.

 Response: The information was added to the manuscript, along with supporting references.

Please see pp. 14 and the additional references in p. 13 and 19.

  1. I totally agree with the authors’ response: "These studies were based on the stability of the sensor after successive readings with redox probe and washings steps with the buffer. Within these conditions, the sensor showed stable electrochemical measurements." to the following issue: "The authors showed that they studied the influence of the cycling potential on the NPPy formation, but they didn’t wrote anything about how / why they selected the number of cycles (10) and the scan rate (20 mV/s)?" but I recommend to specify this briefly in the manuscript

Response: Thank you so much for your comment. Additional information was added to the manuscript. Please check this in p. 9.

  1. Why did the authors change the OX axis of the calibration curves for MPPy and NPPY from log [Cyst, c] to Potential Applied (Figure 6)?????? Please verify and correct!

 Response: The reviewer is correct. This was a mistake. Figure 6 was replaced by a new one.

  1. The authors have specified now in the text that the used human serum from normal individuals was "spiked with specific interfering compounds" but I recommend to list here again the added interfering species (Creatine kinase-MB (CK-MB) 0.2g/L, AA 0.15g/L, Crea 1g/L and BSA 12g/L) which were listed in section 2.2.

Despite the fact that the abbreviations are obvious, I recommend to explain Cre and BSA.

Response: We have added again this information to section 3.6 and deleted the abbreviations throughout, also in section 2.2. Their little use did not justify their existence. This makes it simpler.

Please check p. 5 and 16.
